# The determinants of handwashing behaviour among internally displaced women in two camps in the Kurdistan Region of Iraq

Aso Zangana[1], Nazar Shabila[2], Tom Heath[3], Sian White[4]*

1 Kurdistan Board of Medical Specialties (KBMS), Erbil, Kurdistan, Iraq, 2 Department of Community Medicine, Hawler Medical University, Erbil, Kurdistan, Iraq, 3 Action Contre la Faim, Paris, France, 4 Department of Disease Control, London School of Hygiene and Tropical Medicine, London, United Kingdom

* sian.white@lshtm.ac.uk

## Abstract

### Background

Diarrhoea is one of the most common causes of mortality and morbidity among populations displaced due to conflict. Handwashing with soap has the potential to halve the burden of diarrhoeal diseases in crisis contexts. This study aimed to identify which determinants drive handwashing behaviour in post-conflict, displacement camps.

### Methods

This study was conducted in two camps for internally displaced people in the Kurdistan Region of Iraq. A Barrier Analysis questionnaire was used for assessing the determinants of hand washing behaviour. Participants were screened and classified as either 'doers' (those who wash their hands with soap at critical times) or 'non-doers' (those who do not wash their hands with soap at critical times). Forty-five doers and non-doers were randomly selected from each camp and asked about behavioural determinants. The Barrier Analysis standard tabulation sheet was used for the analysis.

### Results

No differences were observed between doers and non-doers in relation to self-efficacy, action efficacy, the difficulties and benefits of handwashing, and levels of access to soap and water. In the first of the two camps, non-doers found it harder to remember to wash their hands (P = 0.045), had lower perceived vulnerability to diarrhoea (P = 0.037), lower perceived severity of diarrhoea (P = 0.020) and were aware of 'policies' which supported handwashing with soap (P = 0.037). In the second camp non-doers had lower perceived vulnerability to diarrhoea (P = 0.017).

### Conclusions

In these camp settings handwashing behaviour, and the factors that determine it, was relatively homogenous because of the homogeneity of the settings and the socio-demographics

**Data Availability Statement:** The Barrier Analysis Questionnaire used during this study is attached as a supplementary material. All data analysis files are

available from the Figshare database (URL: https://doi.org/10.6084/m9.figshare.8152751.v1)

**Funding:** This reserach was made possible by the generous support of the American people through the United States Agency for International Development (USAID). The contents are the responsibility of the study authors and do not necessarily reflect the views of USAID or the United States Government. A grant from the Office of U.S. Foreign Disaster Assistance was recieved by SW (award number AID-OFDA-G-16-00270). Funder website: https://www.usaid.gov/who-we-are/organization/bureaus/bureau-democracy-conflict-and-humanitarian-assistance/office-us. The funders played no role in the study design, data collection, analysis, decision to publish or preperation of the manuscript.

**Competing interests:** The authors have declared that no competing interests exist.

of population. Handwashing programmes should seek to improve the convenience and quality of handwashing facilities, create cues to trigger handwashing behaviour and increase perceived risk. We identify several ways to improve the validity of the Barrier Analysis method such as using it in combination with other more holistic qualitative tools and revising the statistical analysis.

## Background

During conflicts, children under the age of five are twenty times more likely to die from diarrhoeal diseases rather than as a direct consequence of violence [1]. Handwashing with soap is considered to be one of the most cost-effective public health interventions [2] and has the potential to reduce diarrhoea by 23% to 48% [3–7]. However, the prevalence of handwashing with soap after contact with excreta is estimated to be 19% globally, and prevalence is even lower at other critical times (e.g. before food preparation, before eating, before feeding a child or after cleaning a child's bottom) [8]. Despite the increased risk of diarrheal disease morbidity and mortality among displaced populations [9], handwashing rates remain sub-optimal in the aftermath of crises [10,11].

These low prevalence rates are unlikely to just be due to a lack of knowledge about the health benefits of handwashing. Studies have shown that even in areas of low literacy, populations are well able to explain the link between handwashing and disease avoidance [10,12]. Researchers working in non-emergency settings have identified a range of behavioural determinants likely to affect handwashing with soap. These determinants include the availability of handwashing facilities, soap and water; social norms and support mechanisms; motivations like disgust, nurture (the desire to do what is best for your child) and affiliation (the desire to fit in with a social group); risk perception; self-efficacy; and broader contextual factors [13–17]. In the wake of a humanitarian crisis substantial programmatic attention is given to the promotion of handwashing with soap but often such programmes have been unable to achieve substantial behaviour change [11]. One reason for this may be that there is limited evidence about whether the determinants identified in stable settings are likely to be the same in crises situations.

In stable settings, we are increasingly seeing that hygiene programme designers incorporate a learning phase prior to programme design (often described as 'formative research') [8,12,18–21]. This normally involves programme staff trying to understand the barriers and enablers of behaviour within a specific context. A mix of qualitative and quantitative methods are normally employed. Formative research can span from several weeks to many months and is a relatively resource heavy and high-capacity task. These time and resource demands mean that formative research is often compromised or omitted in humanitarian crises [22,23].

This study aims to contribute to improving our understanding of the determinants of behaviour in humanitarian crises. It does so by exploring barriers to handwashing with soap among women living in two displacement camps in the Kurdistan Region of Iraq (KRI). Through this research we also aim to determine whether existing, rapid methods assessing behavioural determinants are feasible to conduct in crisis settings. As such we have employed the Barrier Analysis approach in this study setting and seek to appraise the strengths and limitations of this tool.

## Methods

### Study site

This study was conducted in Duhok Province during June and July 2017. At this time 3.3 million Iraqis were displaced due to conflict [24]. Two camps for internally displaced persons (IDPs) were purposively selected to reflect different cultures, living conditions, durations of displacement, and different modalities of accessing hygiene infrastructure and products. The first, Nargazliya Camp (henceforth referred to as C1) housed 9,905 people at the time of this research. The population was predominantly Arab from the city of Mosul and its surrounding villages. C1 had been open for about six months at the time of this research and displaced people were still arriving on a daily basis, while others were beginning to return home to their villages. Sheikhan Camp (henceforth referred to as C2) was the other site selected for this research. Its population was more constant. At the time of this research C2 housed 5,371 Yazidi (*Êzidî*) people who had fled from the town of Sinjar and its surrounding villages in the summer of 2014.

Residents of both camps fled from areas which had been taken over by the Islamic State of Iraq and the Levant (ISIS). The nature of this crisis meant that all our research participants had been exposed to extreme violence in the past three years. Through consultations with camp residents and staff we learned that many people within the camps were still experiencing trauma at the time the research was conducted. Camp conditions generally met the SPHERE standards [25] but remained sub-optimal in many other ways. For example, at the time of this research the average temperature in these camps ranged between 45–50˚C. Plastic tents and infrequent access to electricity meant that for most of the day there was no means of keeping cool. C1 was a 'closed camp' meaning that at the time of the research the population were unable to leave without formal permission. All communication equipment (e.g. phones or computers) was taken from C1 residents upon entry to the camp—a measure reportedly taken because of 'security concerns'. Many of the residents had come from urban or peri-urban areas and were used to a relatively high standard of living prior to the conflict. For example, the displaced population would have previously been accustomed to pour-flush toilets and piped water.

In both camps water, sanitation and hygiene (WASH) infrastructure were provided to residents by non-government organisations (NGOs). In C1 WASH facilities were shared between six shelters (about 30 people), while in C2 each family had its own shower, toilet and kitchen. In both camps, water was stored in large tanks and accessible through taps inside the WASH facilities. There were no limitations on the amount of water the IDPs could consume in either camp. At the time of this research hygiene kit distribution (including soap) and hygiene promotion was ongoing in C1. Hygiene promotion was ongoing in C2, however, hygiene kit distribution had ceased and camp residents were responsible for buying their own soap. In both camps hygiene promotion was done by international and local NGOs in conjunction with hygiene promoters from the camp population. In both settings hygiene promotion was done through house-to-house visits. Hygiene promoters taught people a step-by-step process for how hands should be washed and used an image of the F-Diagram to explain faecal-oral disease transmission.

### The barrier analysis method

Barrier Analysis is a standardised rapid assessment tool which is part of the Designing for Behaviour Change Framework [26]. The Barrier Analysis approach is intended to be used in advance of designing a behaviour change programme. It allows programme designers to

identify key barriers and motivators of desirable behaviours (such as handwashing with soap) which can then be used to develop strategies for behaviour change. The Barrier Analysis approach can be considered to be part of a family of approaches which compare the perspectives of people who practice a behaviour ('doers') with those who do not practice the same behaviour ('non-doers'). The RANAS framework, which is widely used in the WASH sector, also uses a doer/non-doer method for understanding behaviour [27]. These approaches are typically grounded in cognitive psychology and are designed with programme implementers in mind. The analysis process resembles that of a case-control study, allowing users to clearly pinpoint the factors that are most likely to enable or inhibit behaviour.

This study used the standardized Barrier Analysis questionnaire [28] for assessing the determinants of handwashing behaviour (S1 File). The Barrier Analysis approach was chosen for this research as it is widely used by the development and humanitarian sectors to inform behaviour change strategy. To date it has reportedly been used by more than 20 NGOs in 50 countries [29]. Despite the common usage of the Barrier Analysis approach, results and reflections on this method are rarely published in peer reviewed journals. Our research team was interested in identifying the strengths and limitations of the Barrier Analysis method and comparing findings with other observational and ethnographical data collection tools (these were implemented subsequently and will be reported elsewhere).

We started by defining the behaviour, the details of when and how this behaviour was to be practiced and priority groups whose behaviour we were interested in (see Table 1). This helped to inform our sampling and survey process. The Barrier Analysis questionnaire consists of two

**Table 1. Table of definitions based on the barrier analysis approach and adapted for this study.**

| Key term | Definition |
|---|---|
| Target Behaviour | Handwashing with soap |
| Priority groups | Mothers of children under the age of five |
| Details of behaviour | Handwashing with water and soap at critical times. |
| | Critical times defined as 1) before preparing food, 2) before eating, 3) before feeding a child, 4) after using the toilet and 5) after cleaning a child's bottom. |
| Perceived self-efficacy | An individual's belief that he/she can wash their hands with soap given his/her current knowledge and skills. |
| Perceived social norms | The perception that people important to an individual think that he/she should wash their hands with soap. |
| Perceived positive consequences | The positive things a person thinks will happen as a result of handwashing with soap. |
| Perceived negative consequences | The negative things a person thinks will happen as a result of handwashing with soap. |
| Access | The availability of the needed products or services (e.g. soap, water, handwashing facilities) required for handwashing with soap. This includes barriers related to the cost, distance, and cultural acceptability of these products and services. |
| Cues to action / reminders | The presence of reminders that help a person remember to wash their hands with soap. |
| Perceived susceptibility | A person's perception of how vulnerable or at risk they are to getting diarrhoea. |
| Perceived vulnerability | The extent to which a person believes that the diarrhoea is a serious illness. |
| Perceived action efficacy | The extent to which a person believes that by practicing handwashing with soap they will be able to avoid getting diarrhoea. |
| Perceived divine will | The extent to which a person believes that it is God's will (or the gods' wills) for him/her to get diarrhoea and/or to overcome it. |
| Policy | The presence of laws and regulations that may affect whether people wash their hands with soap or which affect their access to relevant products and services. |
| Culture | The extent to which local history, customs, lifestyles, values, and practices may affect whether people wash their hands with soap. |

main parts. The first part is designed to classify the participant as either a 'doer' (a person who practices handwashing with soap) or a 'non-doer' (a person who does not practice handwashing with soap). The screening process used a combination of self-reported handwashing behaviour and proxy measures of handwashing behaviour (such as the observed presence of used soap at the handwashing facility). The second part of the questionnaire consisted of closed and open-ended questions exploring the 12 determinants of behaviour change. Specifically, the Barrier Analysis approach explores the following determinants: perceived self-efficacy, perceived social norms, perceived positive consequences, perceived negative consequences, access to products and services, cues to action, perceived susceptibility, and perceived vulnerability, perceived action efficacy, divine will, policy and culture. Table 1 provides a definition of each of these determinants drawn from the Barrier Analysis guidelines [28].

## Enumerator training and questionnaire adaption

The data collection team underwent a three-day training conducted by the last author (SW). This included an overview of behaviour change and the Barrier Analysis questionnaire. The training involved opportunities to role-play using the BA questionnaire in the classroom, prior to piloting in the field sites. The data collection team translated the Barrier Analysis questionnaire into Arabic and Kurdish (Kurmanji). In order to arrive at the most accurate translated terms we used a process of brainstorming synonyms, back-translation and consultations with members of the local population through a focus-group discussion. Prior to the survey we pilot-tested the translated tool with a small number of households in the camps and made some small adjustments to enhance clarity.

## Sampling

The study team administered the questionnaires to women who had a child under the age of five. These women were chosen as the target population because in this region they are the primary caregivers of children and responsible for most household tasks. Participants were selected through random sampling. Maps of both camps were obtained and each block was numbered. Blocks were selected using a random number generator on an Android device. A second random number was generated to select the shelter within the block. When we found a shelter that did not fulfil the criteria, or did not consent to participate, it was excluded, and we selected a neighbouring household by moving in a clockwise direction.

We aimed to select an equal number of doers and non-doers in each camp. The Barrier Analysis approach recommends a sample size of 45 doers and 45 non-doers. This relatively small sample size is argued to be sufficient because the Barrier Analysis method is designed to identify significant differences in behavioural determinants (defined as results with statistical significance of $P < 0.05$) [30]. For this study, 45 doers and 45 non-doers were selected from each camp resulting in a total sample of 180 people. Participants continued to be screened and sampled until these figures were met.

## Data collection and management

Data was collected by a team consisting of two persons, the lead author (AK—male) and a research assistant (female). Both individuals were present in all households to increase the acceptability of the questionnaire process. One person asked the questions while the other acted as a scribe, documenting by hand the key elements of the participant's answer. Both team members spoke Arabic and Kurdish, with the questionnaire being administered in whichever language the participants felt most comfortable in. All responses were entered into an excel spreadsheet on the same day as it was collected to maintain quality and identify any

missing data. If missing data was identified or responses were unclear, this process allowed us to return to the household the next day for clarification.

## Data analysis

The data collection team and the last author classified the qualitative responses thematically, through a collective discussion. At the end of this process we tallied the number of responses in each category, and by their doer or non-doer classifications. These figures where then entered into the standardised Barrier Analysis tabulation sheet to draw conclusions from the data. This allows for closed-answer, quantitative data to be easily summarized and compared using the standard Barrier Analysis approach involving Chi-square tests and the generation of an odds ratio. The Barrier Analysis tabulation sheet highlights differences between doers and non-doers based on P values of ≤0.05.

## Ethics

Informed written consent was obtained from each participant. The research was approved by the Ethics committees at the London School of Hygiene and Tropical Medicine and Hawler Medical University. Permission to work in both camps was provided by the Board of Relief and Humanitarian Affairs in Kurdistan and all non-government organisations in the camp were informed of our work.

## Results

### Classification of doers and non-doers

To be classified as a doer, participants had to mention at least three of the five critical hand-washing times when asked 'yesterday, what were all the moments that you washed your hands?'. They also had to report that they used soap when handwashing and had to have a used bar of soap present at the handwashing facility (based on a spot-check by the data collection team).

The most commonly reported 'moment' for handwashing with soap was before preparing food (number reporting this = 154/180). Doers in both camps were observed to have a used bar of soap near WASH facilities (in the kitchen or near the latrine). Only six non-doer households were found to not have soap. The majority of non-doers were found to keep their soap elsewhere in the house.

### Perceived self-efficacy

Across both camps, all the doers felt that they were able to wash their hands with soap at the five critical times given their current knowledge, skills and their available resources. Most non-doers also reported feeling able to wash their hands at critical times (C1 = 96%, C2 = 98%).

When asked about factors that made handwashing easier, there was a high level of consistency between doers and non-doers and across the two camps. All the factors mentioned by participants were related to the availability and close proximity of resources such as piped water, soap and handwashing facilities (see Table 2). Participants in C2 were less likely than participants in C1 to mention that handwashing stations and soap increased their ease of handwashing (p = 0.002).

In both camps, there were a variety of difficulties which prevented mothers from sometimes washing their hands (Table 3). However, there were no substantial differences in the difficulties reported by doers and non-doers.

**Table 2. Comparison of the doers and non-doers in the two camps regarding factors that make it easier to wash hands with soap.**

| | Camp 1 | | | | | Camp 2 | | | | |
|---|---|---|---|---|---|---|---|---|---|---|
| Participant Responses | Doers | Non-Doers | Difference | Odds ratio | P value | Doers | Non-Doers | Difference | Odds ratio | P value |
| **What makes it easier for you to wash your hands with soap at the five critical times each day?** | | | | | | | | | | |
| Availability of piped water | 44 (98%) | 43 (96%) | 2% | 2.05 | 0.500 | 45 (100%) | 43 (96%) | 4% | | 0.247 |
| Handwashing facilities are available | 20 (44%) | 18 (40%) | 4% | 1.20 | 0.416 | 10 (22%) | 9 (20%) | 2% | 1.14 | 0.500 |
| Close proximity of handwashing facilities | 6 (13%) | 3 (7%) | 7% | 2.15 | 0.242 | 3 (7%) | 3 (7%) | 0% | 1.00 | 0.662 |
| Soap is available | 41 (91%) | 38 (84%) | 7% | 1.89 | 0.261 | 11 (24%) | 9 (20%) | 4% | 1.29 | 0.400 |

There were no significant differences in the difficulties mentioned by doers and non-doers in relation to handwashing. Participants in C2 typically listed a greater number of difficulties than participants in C1. In both camps participants reported difficulties related to the hot weather, the cleanliness of the broader environment, a lack of privacy and mental health challenges. Some difficulties were more pronounced in C1. For example, participants reported that the water for handwashing was hot, the handwashing facilities were shared and too far away, and that the living environment was overcrowded. In contrast, the issues predominately reported in C2 included the quantity and quality of water, a lack of space in bathrooms and kitchens, and broken or damaged handwashing facilities.

## Perceived positive consequences

Participants cited many positive consequences of handwashing (see Table 4). The majority of women in both sites said that the main positive consequence of handwashing with soap was the removal of dirt and the prevention of disease. In C1 both of these beliefs were actually more common among non-doers. For example, non-doers were 18% more likely than doers to

**Table 3. Difficulties which hinders the mothers from washing their hands for both doers and non-doers in the two camps.**

| | Camp 1 | | | | | Camp 2 | | | | |
|---|---|---|---|---|---|---|---|---|---|---|
| Participant Responses | Doers | Non-Doers | Difference | Odds ratio | P value | Doers | Non-Doers | Difference | Odds ratio | P value |
| **What makes it difficult for you to wash your hands with soap at the five critical times each day?** | | | | | | | | | | |
| The environment is dirty and uncomfortable | 13 (29%) | 7 (16%) | 13% | 2.21 | 0.102 | 9 (20%) | 16 (36%) | -16% | 0.45 | 0.079 |
| Hot weather and lack of electricity cause people to be sweaty | 9 (20%) | 11 (24%) | -4% | 0.77 | 0.400 | 26 (58%) | 25 (56%) | 2% | 1.09 | 0.500 |
| Soap is unavailable or affordable | 8 (8%) | 10 (22%) | -4% | 0.76 | 0.396 | 21 (47%) | 25 (56%) | -9% | 0.70 | 0.264 |
| Quality of water is poor | 0 (0%) | 1 (2%) | -2% | 0.00 | 0.500 | 25 (56%) | 21 (47%) | 9% | 1.43 | 0.264 |
| There is not enough water | 1 (2%) | 0 (0%) | 2% | | 0.500 | 5 (11%) | 6 (13%) | -2% | 0.81 | 0.500 |
| Not enough space in the bathroom and the kitchen | 1 (2%) | 0 (0%) | 2% | | 0.500 | 4 (9%) | 6 (13%) | -4% | 0.63 | 0.370 |
| Poor design of the handwashing facilities | 2 (4%) | 6 (13%) | -9% | 0.30 | 0.133 | 2 (4%) | 1 (2%) | 2% | 2.05 | 0.500 |
| Our handwashing facilities are shared | 1 (2%) | 4 (9%) | -7% | 0.23 | 0.180 | 0 (0%) | 0 (0%) | 0% | - | 1.000 |
| The water is hot | 4 (9%) | 5 (11%) | -2% | 0.78 | 0.500 | 0 (0%) | 0 (0%) | 0% | | 1.000 |
| There is no privacy | 3 (7%) | 2 (4%) | 2% | 1.54 | 0.500 | 4 (9%) | 2 (4%) | 4% | 2.10 | 0.338 |
| The living environments are overcrowded | 1 (2%) | 4 (9%) | -7% | 0.23 | 0.180 | 0 (0%) | 0 (0%) | 0% | | 1.000 |
| The handwashing facilities are far away | 2 (4%) | 2 (4%) | 0% | 1.00 | 0.692 | 0 (0%) | 1 (2%) | -2% | 0.00 | 0.500 |
| Hand washing facilities are damaged or broken. | 0 (0%) | 0 (0%) | 0% | 1.54 | 0.500 | 0 (0%) | 4 (9%) | -9% | 0.00 | 0.058 |
| Mental health challenges | 1 (2%) | 1 (2%) | 0% | 1.00 | 0.753 | 0 (0%) | 2 (4%) | -4% | 0.00 | 0.247 |

**Table 4. Comparison of the responses of doers and non-doers in each camp regarding the positive consequences of handwashing.**

| Participant Responses | Camp 1 | | | | | Camp 2 | | | | |
|---|---|---|---|---|---|---|---|---|---|---|
| | Doers | Non-Doers | Difference | Odds ratio | P value | Doers | Non-Doers | Difference | Odds ratio | P value |
| **What are the advantages of washing your hands with soap at the five critical times each day?** | | | | | | | | | | |
| To get rid of dirtiness | 30 (67%) | 38 (84%) | -18% | 0.37 | 0.042* | 42 (93%) | 42 (93%) | 0% | 1.00 | 0.662 |
| To get rid of germs and disease | 39 (87%) | 37 (82%) | 4% | 1.41 | 0.386 | 40 (89%) | 38 (84%) | 4% | 1.47 | 0.379 |
| To feel more relaxed psychologically | 11 (24%) | 10 (22%) | 2% | 1.13 | 0.500 | 7 (16%) | 5 (11%) | 4% | 1.47 | 0.379 |
| To prevent food from being contaminated | 2 (4%) | 5 (11%) | -7% | 0.37 | 0.217 | 1 (2%) | 0 (0%) | 2% | 0.00 | 0.500 |
| To look and smell good or improve my personal image | 1 (2%) | 2 (4%) | -2% | 0.49 | 0.500 | 0 (0%) | 3 (7%) | -7% | 0.00 | 0.121 |
| To improve my child's health | 2 (4%) | 4 (9%) | -4% | 0.48 | 0.338 | 0 (0%) | 1 (2%) | -2% | 0.00 | 0.500 |
| To prevent insects, lice and flies | 1 (2%) | 0 (0%) | 2% | 2.05 | 0.500 | 2 (4%) | 1 (2%) | 2% | 2.05 | 0.500 |

report that getting rid of dirt was a key advantage of handwashing (p = 0.042). The third most commonly mentioned benefit was that handwashing could contribute to feeling more psychologically relaxed. Women also said that handwashing allows them to keep their children healthy and protected from disease and that it helps them feel more attractive.

**Perceived negative consequences.** The majority of women in both camps did not think that there were any negative consequences of handwashing with soap. In C2, non-doers were 18% more likely than doers to report that they did not face any negative consequences from handwashing with soap (doers = 80%, non-doers = 98%, p = 0.008) while in C1 the reverse was true with doers 9% more likely to perceive there to be no negative consequences of handwashing (doers = 91%, non-doers = 82%, p = 0.176). The negative consequences related to dermatological consequences, with a total of 15 people across both sites reporting cracked or irritated hands and one other person feeling that handwashing caused their skin to become lighter in colour.

**Social norms.** In general, mothers in both sites reported that the people around them approved of them washing their hands with water and soap at the five critical times. However, a total of 18 people (20%) across both sites were not sure what other people thought about handwashing and 25 others (28%) thought people disapproved of regular handwashing with soap. In C1, 17 participants(38%) felt that their neighbours sometimes disapproved of them regularly washing their hands, while only one person (2%) shared this belief in C2. Doers in C1 appeared to receive substantial support from their mothers, with doers being 16% more likely to report this than non-doers (p-value = 0.015). In both camps, most of the mothers said that they relied on their own motivation to wash their hands, rather than the social approval of others. Table 5 describes the participants' responses on social norms.

## Perceived access

In both camps, the majority of participants said that accessing sufficient soap and water for handwashing was somewhat difficult or very difficult (Table 6), with residents of C2 (65 people in C2 compared to 39 in C1) and non-doers (p-value C1 = 0.76, p-value C2 = 0.90) being more likely to report difficulty.

## Cues to action

In both camps, non-doers were more likely than doers to report that it was sometimes difficult to remember to wash their hands with water and soap at the five critical times (p-value

**Table 5. Comparison of the doers and non-doers in each camp regarding perceived social norms.**

| Participant Responses | Camp 1 | | | | | Camp 2 | | | | |
|---|---|---|---|---|---|---|---|---|---|---|
| | Doers | Non-Doers | Difference | Odds ratio | P value | Doers | Non-Doers | Difference | Odds ratio | P value |
| **Who are the people that would approve of you washing your hands with soap at the five critical times each day?** | | | | | | | | | | |
| I do it for myself | 39 (87%) | 41 (91%) | -4% | 0.63 | 0.370 | 42 (93%) | 42 (93%) | 0% | 1.00 | 0.662 |
| My mother | 8 (18%) | 1 (2%) | 16% | 9.51 | 0.015* | 1 (2%) | 1 (2%) | 0% | 1.00 | 0.753 |
| My husband | 6 (13%) | 9 (20%) | -7% | 0.62 | 0.286 | 5 (11%) | 1 (2%) | 9% | 5.50 | 0.101 |
| Religious leaders | 0 (0%) | 2 (4%) | -4% | 0.00 | 0.247 | 0 (0%) | 0 (0%) | 0% | | 1.000 |
| **Who are the people that would disapprove of you washing your hands with soap at the five critical times each day?** | | | | | | | | | | |
| No one | 35 (78%) | 33 (73%) | 4% | 1.27 | 0.403 | 44 (98%) | 45 (100%) | -2% | 0.00 | 0.500 |
| Neighbours | 7 (16%) | 10 (22%) | -7% | 0.64 | 0.296 | 1 (2%) | 0 (0%) | 2% | 0.00 | 0.500 |

C1 = 0.045, p-value C2 = 0.204). However, most of the mothers experienced no difficulty with remembering to wash their hands as shown in Table 7.

## Perceived risk

Table 8 describes participant perceptions of perceived vulnerability to diarrhoea, perceived severity of diarrhoea and the action efficacy of handwashing. Participants in C1 perceived themselves to be at much greater risk of diarrhoea than participants in C2, with 36 women in C1 reporting that they felt that their child was likely to get diarrhoea in the next three months, compared to just 12 in C2. Doers in both camps were also more likely to perceive their children as being susceptible to diarrhoea. For example, doers in C1 were 2.94 times more likely than non-doers to say that it was 'somewhat likely' that their children would get diarrhoea in the coming months (p-value = 0.037), while non-doers in C2 were 2.7 times more likely than doers to think that it was not at all likely that their children would get diarrhoea (p-value = 0.017). In C1 most doers felt that diarrhoea was a 'very serious problem' and were 2.92 times more likely to give this response when compared with non-doers (p-value = 0.02). In C2 this difference was not observed. The perceived action efficacy was high in C2 with both doers and non-doers believing that handwashing with soap at critical times can prevent diarrhoea (83% overall). It was considerably lower in C1 (61% overall) and in this camp doers were more likely to doubt the action efficacy of handwashing against diarrhoea.

## Religion, culture and policy

In both camps, no significant difference existed between the doers and non-doers regarding religion, culture and policy. The vast majority of participants in both camps did not believe that it was 'God's will' that determined whether children got diarrhoea (94% in C1 and 92% in C2, p = 0.5 in both camps). All participants in both camps said that there were no cultural taboos that prevented handwashing. In C1 non-doers were more likely to report that there

**Table 6. Comparison of the doers and non-doers in each camp regarding the perceived access to soap and water.**

| Participant Responses | Camp 1 | | | | | Camp 2 | | | | |
|---|---|---|---|---|---|---|---|---|---|---|
| | Doers | Non-Doers | Difference | Odds ratio | P value | Doers | Non-Doers | Difference | Odds ratio | P value |
| **How difficult is it to get the soap and water you need to wash your hands at the five critical times each day?** | | | | | | | | | | |
| Very difficult | 18 (40%) | 21 (47%) | -7% | 0.76 | 0.335 | 32 (71%) | 33 (73%) | -2% | 0.90 | 0.500 |
| Somewhat difficult | 15 (33%) | 13 (29%) | 4% | 1.23 | 0.410 | 10 (22%) | 11 (24%) | -2% | 0.88 | 0.500 |
| Not difficult at all | 11 (24%) | 10 (22%) | 2% | 1.13 | 0.500 | 2 (4%) | 1 (2%) | 2% | 2.05 | 0.500 |

**Table 7. Comparison of the doers and non-doers in each camp regarding the cues to action.**

| | Camp 1 | | | | | Camp 2 | | | | |
|---|---|---|---|---|---|---|---|---|---|---|
| Participant Responses | Doers | Non-Doers | Difference | Odds ratio | P value | Doers | Non-Doers | Difference | Odds ratio | P value |
| How difficult is it to remember to wash your hands with soap at the five critical times each day? | | | | | | | | | | |
| Very difficult | 1 (2%) | 0 (0%) | 2% | | 0.500 | 0 (0%) | 1 (2%) | -2% | 0.00 | 0.500 |
| Somewhat difficult | 2 (4%) | 8 (18%) | -13% | 0.22 | 0.045* | 6 (13%) | 10 (22%) | -9% | 0.54 | 0.204 |
| Not difficult at all | 42 (93%) | 37 (82%) | 11% | 3.03 | 0.098 | 39 (87%) | 34 (76%) | 11% | 2.10 | 0.141 |

were community laws or rules in place to encourage handwashing (doers = 31, non-doers = 39, p-value = 0.037). Specifically, they referred to the role of non-governmental organizations in promoting handwashing. Doers in C1 were 2.4 times more likely to report that no such rules existed (p-value = 0.037). In C2 there were no significant differences between doers and non-doers; however, participants in this camp were more likely to report the absence of any community rules (rules present = 40%, rules absent = 60%).

## Discussion

This study used the Barrier Analysis method to explore the determinants affecting handwashing with soap among IDP populations in two camps in KRI. Here we summarise the findings according to the classification of doers and non-doers and compare behaviour in the two camps. We also reflect on the Barrier Analysis method, highlighting the strengths and weaknesses of the approach.

### Summary of the findings

Our study identified a surprising level of homogeneity between the reported behaviour, beliefs and perceptions of doers and non-doers in relation to handwashing with soap. Doers and non-doers both felt able to wash their hands (self-efficacy) and believed that it would prevent them getting diarrhoea (action efficacy). Both groups believed that religion and culture had minimal effects on handwashing and both groups described similar difficulties, benefits, and levels of access to soap and water. These similarities are likely to be a reflection of the fact that the populations and physical environment within each camp were homogeneous.

**Table 8. Comparison of the doers and non-doers in each camp regarding the perceived risk.**

| | Camp 1 | | | | | Camp 2 | | | | |
|---|---|---|---|---|---|---|---|---|---|---|
| Participant Responses | Doers | Non-Doers | Difference | Odds ratio | P value | Doers | Non-Doers | Difference | Odds ratio | P value |
| How likely is it that your child will get diarrhoea in the coming three months? | | | | | | | | | | |
| Very likely | 17 (38%) | 19 (42%) | -4% | 0.83 | 0.415 | 8 (18%) | 4 (9%) | 9% | 2.22 | 0.176 |
| Somewhat likely | 14 (31%) | 6 (13%) | 18% | 2.94 | 0.037* | 18 (40%) | 11 (24%) | 16% | 2.06 | 0.088 |
| Not likely at all | 14 (31%) | 20 (44%) | -13% | 0.56 | 0.138 | 19 (42%) | 30 (67%) | -24% | 0.37 | 0.017* |
| How serious would it be if your child got diarrhoea? | | | | | | | | | | |
| Very serious problem | 36 (80%) | 26 (58%) | 22% | 2.92 | 0.020* | 33 (73%) | 34 (76%) | -2% | 0.89 | 0.500 |
| Somewhat serious problem | 5 (11%) | 11 (24%) | -13% | 0.39 | 0.083 | 9 (20%) | 5 (11%) | 9% | 2.00 | 0.192 |
| Not serious at all | 4 (9%) | 6 (13%) | -4% | 0.63 | 0.370 | 3 (7%) | 6 (13%) | -7% | 0.46 | 0.242 |
| How likely is it that your child will suffer from diarrhoea if you wash your hands with soap at the five critical times each day? | | | | | | | | | | |
| Very likely | 4 (9%) | 0 (0%) | 9% | | 0.058 | 0 (0%) | 1 (2%) | -2% | 0.00 | 0.500 |
| Somewhat likely | 17 (38%) | 13 (29%) | 9% | 1.49 | 0.251 | 7 (16%) | 7 (16%) | 0% | 1.00 | 0.614 |
| Not likely at all | 24 (53%) | 31 (69%) | -16% | 0.52 | 0.097 | 38 (84%) | 37 (82%) | 2% | 1.17 | 0.500 |

Generally, participants across both camps felt that there were minimal negative consequences of handwashing. However, doers in C1 were more likely to report skin irritations, while in C2 this was more common among non-doers. Participants cited a range of benefits associated with handwashing but interestingly non-doers, particularly in C1, were more likely to report that the primary benefit was the removal of dirt from hands (p-value C1 = 0.042). One possible explanation for this finding is that non-doers may be more likely to reactively wash their hands when hands are visibly dirty rather than at critical times. There is evidence from others studies about visible dirt acting as a key motivator for handwashing with soap. [12,31]

Most participants said that they were self-motivated to wash their hands and did not require support from others. However, doers in C1 were more likely than non-doers to receive social approval from their mothers to practice handwashing with soap (p-value = 0.015). This finding was not replicated in C2. Most participants said they found handwashing easy to remember. However, non-doers in both camps were more likely to report challenges remembering to always wash their hands with soap at critical times. This finding was particularly pronounced in C1 (p-value = 0.045). Doers were more likely to feel that their children were susceptible to diarrhoea (p-value C1 = 0.037, p-value C2 = 0.017). Doers in C1 were more likely than non-doers to describe diarrhoea as a 'very serious problem' (p-value = 0.02), but no such difference was observed in C2. Doers in C1 were more aware of 'policies' which supported handwashing with soap, specifically citing the role of non-governmental organizations in promoting handwashing (p-value = 0.037). No such difference was observed in C2.

Several of our findings may at first seem to run counter to logical assumptions about behaviour. For example, in C2 non-doers were more likely to report that there were no negative consequences to handwashing. One explanation for this finding is that since non-doers wash their hands less frequently they may have also not encountered some of the negative consequences that doers reported (e.g. skin irritation). In C1 doers doubted the action efficacy of handwashing more than non-doers. One explanation for this finding might be that doers, as regular hand-washers, realise that handwashing is important but not sufficient to block all routes of diarrhoeal disease transmission. Alternatively, it may be that these findings occurred by chance.

The similarity of the findings is interesting given that the populations in the two camps were quite different–people came from different geographical locations, were from different cultures, had different religions and had been displaced for different periods of time. There was also a difference in the quality of WASH services provided in the two camps, with C2 having objectively better conditions (namely because WASH facilities were not shared). Despite having objectively better conditions, participants in C2 reported a greater number of barriers to handwashing. This may be because at the acute stage of a crisis (as in C1) people are relieved to receive basic WASH provisions. However, when populations are displaced for an extended period of time (as in C2) they begin to tire of WASH conditions that are substantially poorer than what they were accustomed to prior to displacement. Overall there were more pronounced differences between doers and non-doers in C1 than in C2. This may indicate that camp environments tend to create new emergent norms [32]. That is to say that when people live in condensed living environments for an extended period of time, their behaviour and beliefs become more similar.

The findings highlighted in this study are not dissimilar to studies which have explored the determinants of handwashing behaviour in non-emergency settings. However, there are a few notable exceptions to this. In both camps, the trauma experienced by the populations appeared to affect their behaviour. Some people said that their mental health impaired their ability to wash their hands with soap while others said that handwashing helped them to 'feel more

relaxed psychologically'. Studies in this region have estimated that almost all Yazidi survivors exhibit symptoms of psychiatric disorders [33–35]. Anecdotal evidence indicates rates are likely to be similarly high among Arabs displaced from Mosul [36,37]. It is likely that mental health may be a factor that influences handwashing behaviour in other crisis-affected contexts yet this was unable to be sufficiently documented through the Barrier Analysis method since there were no specific questions exploring this.

Secondly, our findings suggest that people in displacement camps may be more likely to attribute handwashing challenges to factors in the external environment, beyond their control. When asked about handwashing difficulties, people reported being disgusted by the camp environment, describing it as 'dirty,' 'overcrowded' and 'uncomfortable.' They also described feeling motivated to wash their hands because of their increased sweatiness and exposure to the summer heat (they were used to hot temperatures prior to displacement but were now much more directly affected by the weather due to living in tented shelters). People were also dissatisfied with the quality of WASH services in the camp. Frustration with the distance to facilities and the appropriateness of the design of handwashing facilities is likely to be less commonly reported in non-emergency situations where populations are responsible for purchasing and building their own handwashing stations.

Our findings suggest that behavioural interventions targeted at IDPs within these contexts should try to increase perceived social support for handwashing, provide cues to trigger behaviour, and increase perceived risk in relation to both susceptibility and severity. Providing a more dermatologically-friendly soap might help to reduce the perceived negative consequences of handwashing. Improving the design and location of handwashing facilities so that they are more acceptable and convenient is likely to reduce perceived barriers to handwashing practice. Improving handwashing facilities [38–41] and adding behavioural cues [42–44] has been demonstrated to work in other studies in stable settings. Increasing risk should be done with care so as not to create unintended consequences [45]. There is some evidence from other crises that heightening fear only has short term benefits on handwashing behaviour [14,46].

### Reflections on the barrier analysis approach

The Barrier Analysis approach proved feasible to do in an emergency context as it was conducted in both sites, in 14 days, by two staff. The appeal of the approach to practitioners is its ability to translate qualitative responses into quantitative data. Its reliance on 'statistically significant' differences helps practitioners who are new to the field of behaviour change to pinpoint which barriers to focus on.

However, in this study it was this perceived strength, that limited the generation of meaningful insights about behaviour. The standard Barrier Analysis approach is perhaps less suited to settings with high homogeneity (both in terms of population characteristics and the physical settings/access to resources) or where handwashing rates are already relatively high. This is because it is powered to detect major differences in the determinants of behaviour. Our results indicate that in Middle-Eastern camp settings differences between doers and non-doers are likely to be more subtle.

We followed the statistical analysis process recommended by the Barrier Analysis approach. However, we feel there are several limitations of this. Firstly, we feel that Fisher's exact test may be more appropriate than a Pearson chi-square test because of the small sample sizes recommended for Barrier Analysis surveys [47]. Secondly, some of the standard Barrier Analysis question collect ordinal data (See Tables 6,7 and 8). It would be more appropriate to use a Kendall rank correlation coefficient to assess these questions where there are two ordinal-scaled

variables[48]. Even with limitations of the statistical methods recommended by the Barrier Analysis method, there were relatively few 'statistically significant' differences between doers and non-doers in our results. A standard analysis of these results would suggest that there were minimal changes that needed to be made to improve handwashing behaviour in this context. The reliance of the Barrier Analysis method on 'statistically significant' results is also inconsistent with current thinking on statistical interpretation [49] and may down-play the value of the full set of open-ended responses which in this case were rich, varied and programmatically relevant.

We may have observed minimal differences between doers and non-doers because this population was highly exposed to hygiene promotion activities, therefore their responses to self-reported questions may have been affected by social desirability bias. This is a widely recognised limitation of self-reported measures of assessing handwashing behaviour [50,51]. This potential bias, further justifies the need to combine the Barrier Analysis with other methods for exploring behaviour such as proxy measures, monitors, sticker diaries, observation or script-based covert recall [50,52,53]. It is also possible that Barrier Analyses are more appropriate for behaviours where there is a clear way of measuring whether people are doers and non-doers (such as smoking cessation [54]). For a routine behaviour like handwashing with soap, the dichotomy between doers and non-doers may be false—with any given individual remembering to practice on some critical occasions and not on others.

We also found that the questions relating to norms, religion, culture and policy were too narrow, given that they are each assessed with a single closed answer question. We feel that this may have prevented deeper learning about these topics, which are likely to be even more critical in crisis contexts. Future application of Barrier Analyses in conflict-affected settings might consider additional questions on these topics and drawing on a broader literature of norms assessment [55,56].

During our surveys, people often wanted to talk about topics other than handwashing. People often answered the set questions but then went on to share their experiences of the conflict or discuss the broader challenges they faced in the camp. These patterns in participant responses raise some ethical concerns about the appropriateness of very narrow assessment tools in crisis-affected contexts. While Barrier Analysis provides a feasible, rapid way of assessing behaviours that are of interest to public health practitioners, these behaviours may be of relatively low priority to crisis-affected populations given their current predicament. If multiple, similar types of assessments were to be done, as they often are in a crisis, this may cause crisis-affected populations to develop a sense of frustration with the humanitarian system. If others are planning to use the Barrier Analysis approach, then they should plan to locate the method within a broader community dialogue and have in place referral mechanisms to address unanticipated topics that may arise while conducting the questionnaire.

Our study may have been limited by the fact that all 180 questionnaires were conducted by just two people. Larger teams are typically involved in the Barrier Analysis data collection and coding process. While we do not feel this substantially affected the data quality, a larger team may have reduced interviewer fatigue and lead to a richer discussion between team members during the thematic analysis.

## Conclusion

Implementing the Barrier Analysis approach in post-conflict, camp settings was feasible and highlighted some behavioural barriers that could be addressed through hygiene programming. The homogeneity of our results, within and between the two camps, may indicate that routine behaviours like handwashing tend to vary less in camp settings where populations have been

through similar experiences and have access to the same physical infrastructure. Future work in camp-based, post-conflict settings could benefit from combining rapid assessment tools like Barrier Analysis with other more holistic qualitative methods that rely less on self-reported behaviour and which are more sensitive to the diverse needs of displaced people.

## Supporting information

**S1 File. Barrier analysis questionnaire.**
(DOCX)

## Acknowledgments

We would like to thank Basima Ahmed who assisted with the data collection and the team at Action Contre la Faim in Iraq who provided substantial support in terms of logistics, security and office space. We would also like to thank all the people who participated in this research and gave generously of their time.

## Author Contributions

**Conceptualization:** Nazar Shabila, Tom Heath, Sian White.

**Data curation:** Aso Zangana.

**Formal analysis:** Aso Zangana, Sian White.

**Funding acquisition:** Tom Heath, Sian White.

**Investigation:** Aso Zangana.

**Methodology:** Tom Heath, Sian White.

**Project administration:** Nazar Shabila, Tom Heath.

**Resources:** Sian White.

**Supervision:** Sian White.

**Validation:** Sian White.

**Writing – original draft:** Aso Zangana.

**Writing – review & editing:** Nazar Shabila, Tom Heath, Sian White.

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
