## [Decision Letter · Decision Letter 0]

22 Jan 2020

PONE-D-19-25323

The determinants of handwashing behaviour among internally displaced people in two camps in the Kurdistan Region of Iraq

PLOS ONE

Dear Ms White,

Thank you for submitting your manuscript to PLOS ONE. After careful consideration, we feel that it has merit but does not fully meet PLOS ONE’s publication criteria as it currently stands. Therefore, we invite you to submit a revised version of the manuscript that addresses the points raised during the review process.

I am returning your manuscript with three reviews. As you will see, all three reviewers highlight the need for research of this type to be carried out. However, all three express concerns regarding the methodology - specifically the identification and classification of "doers" and "non-doers", the data analysis - specifically the statistical tests used and the interpretation of the data. More detail regarding the camps (e.g. selection process; condition and NGO activities) and how this may have impacted behaviour is also requested. The concept of "emergent norms" is briefly discussed; was the length of time each mother had lived within the camp documented - did this differ between "doers" and "non-doers"? 

Whilst major revision of your manuscript is required, the reviewers acknowledge that studies on handwashing behaviour in humanitarian settings are lacking and so I encourage you to give their comments and suggestions due consideration. Please note, that on resubmission (if you choose to do so), the manuscript will have to go through the second round of review including if deemed necessary, a statistical review.

We would appreciate receiving your revised manuscript by Mar 07 2020 11:59PM. To enhance the reproducibility of your results, we recommend that if applicable you deposit your laboratory protocols in protocols.io, where a protocol can be assigned its own identifier (DOI) such that it can be cited independently in the future. For instructions see: http://journals.plos.org/plosone/s/submission-guidelines#loc-laboratory-protocols

We look forward to receiving your revised manuscript.

Kind regards,

Ginny Moore

Academic Editor

PLOS ONE

Journal Requirements:

2. Please include additional information regarding the survey or questionnaire used in the study and ensure that you have provided sufficient details that others could replicate the analyses. For instance, if you developed a questionnaire as part of this study and it is not under a copyright more restrictive than CC-BY, please include a copy, in both the original language and English, as Supporting Information. Additionally, please include upon how many participants the pre-testing of the questionnaire occurred.

3. Please refer to any post-hoc corrections for multiple comparisons you made during your statistical analyses. If these were not performed please justify why.

4. We note you have included a table to which you do not refer in the text of your manuscript. Please ensure that you refer to Table 7 in your text; if accepted, production will need this reference to link the reader to the Table.

5. Please include captions for your Supporting Information files at the end of your manuscript, and update any in-text citations to match accordingly. Please see our Supporting Information guidelines for more information: http://journals.plos.org/plosone/s/supporting-information

6. Your ethics statement must appear in the Methods section of your manuscript. If your ethics statement is written in any section besides the Methods, please move it to the Methods section and delete it from any other section. Please also ensure that your ethics statement is included in your manuscript, as the ethics section of your online submission will not be published alongside your manuscript.

Reviewers' comments:

Reviewer's Responses to Questions

**Comments to the Author**

1. Is the manuscript technically sound, and do the data support the conclusions?

Reviewer #1: Partly

Reviewer #2: Yes

Reviewer #3: Yes

2. Has the statistical analysis been performed appropriately and rigorously? 

Reviewer #1: No

Reviewer #2: Yes

Reviewer #3: I Don't Know

3. Have the authors made all data underlying the findings in their manuscript fully available?

Reviewer #1: Yes

Reviewer #2: Yes

Reviewer #3: Yes

4. Is the manuscript presented in an intelligible fashion and written in standard English?

Reviewer #1: Yes

Reviewer #2: Yes

Reviewer #3: Yes

5. Review Comments to the Author

Reviewer #1: TECHNICAL FEEDBACK

1. It is good to see a study on handwashing in humanitarian settings. The authors are well justified in their contention that this kind of research is needed.

2. The major technical limitation of the study lies with the identification of doers and non-doers. The authors report that “The screening process used a combination of self-reported handwashing behaviour and proxy measures of handwashing behaviour (such as the observed presence of used soap at the handwashing facility).” Self-reported handwashing is notoriously overreported:

https://journals.lww.com/jhqonline/Abstract/2007/07000/Reliability_and_Validity_of_Hand_Hygiene_Measures.5.aspx

Even direct observation of handwashing can overestimate actual handwashing: http://www.ajtmh.org/content/journals/10.4269/ajtmh.2010.09-0763

Script-based covert recall may be an alternative: https://journals.plos.org/plosone/article?id=10.1371/journal.pone.0136445

In this setting, proxy measures themselves may be inaccurate, and soap may be more available in a humanitarian setting than it would be normally.

Therefore, the study does not convincingly measure handwashing in a valid manner, and this throws into doubt the validity of classification of study participants as doers and non-doers.

3. “Perceived social norms - The perception that people important to an individual think that he/she should wash their hands with soap. Note that in theories such as the Focus Theory of Normative Conduct, this is the definition of injunctive norms. See: Cialdini, R.B.; Reno, R.R.; Kallgren, C.A. (1990). "A focus theory of normative conduct: Recycling the concept of norms to reduce littering in public places". Journal of Personality and Social Psychology. 58 (6): 1015–1026. doi:10.1037/0022-3514.58.6.1015.

4. Wrong statistical methods. In Tables like Table 8, the outcome or dependent variable “How likely is it that your child will get diarrhoea in the coming three months?” is an ordinal variable (See https://en.wikipedia.org/wiki/Ordinal_data for a definition) with three ordered responses. Appropriate measures of correlation include Kendall’s Tau-B, see https://en.wikipedia.org/wiki/Ordinal_data#Bivariate_statistics. It is inappropriate and incorrect to calculate a p-value for each row of the table. One single p-value and measure of association should be calculated with a statistical test like Kendall’s Tau-B. The same applies to Table 6 and 7.

5. In the absence of accompanying qualitative interview and observation data, it is difficult to draw any conclusions about why people do or do not wash their hands.

MINOR SPELLING AND GRAMMAR ISSUES

1. Please do a spell-check e.g. there are three errors here on the first page: “This [research] was made possible by the generous support of the American people through the United States Agency for International Development (USAID). The contents are the responsibility of the study authors and do not necessarily reflect the views of USAID or the United States Government. A grant from the Office of U.S. Foreign Disaster Assistance was [received] by SW (award number AID-OFDA-G-16- 00270). Funder website: https://www.usaid.gov/who-we- are/organization/bureaus/bureau-democracy-conflict-and-humanitarian- assistance/office-us. The funders played no role in the study design, data collection, analysis, decision to publish or [preparation] of the manuscript.”

2. Line 289 – delete ‘of’ or reword that part of sentence

3. Line 388 – change subtler to more subtle

4. Line 397 – remove comma

Reviewer #2: Thank you for this interesting work. Overall, this is well written and useful for the humanitarian sector.

A good edit of the intro, methods, and results is needed; I thought the discussion was particularly well written and organized.

Something that I felt was missing was the discussion of WASH activities in the camps and previous/current hygiene promotion. NGO presence and activities need to be explained. By not describing NGO activities, the assumption is that NGO activities, promotion or distribution of materials or cash/vouchers have no impact on actions - I believe that to be rather significant and must be addressed.

Differences were mentioned briefly about the camps, but more analysis should be presented. If the different camps are actually similar, would combining the data sets provide a better opportunity for significant results.

How the camps were selected from what I would assume to be be dozens of camps needs to be explained.

Within the results there were several interesting findings (significant or not) that would be against logical assumptions, i.e. non-doers thought washing their hands would make them safer compared to doers (or something along those lines). While not significant, some explanation should be given or rationalized, especially since it seemed to occur several times.

Detailed comments:

Line 52-3: Strong statement. Please reference. Limited water availability or lack of soap could also be reasons. 

Line 64: Please be explicit about 'this research gap'

Methods: Please explain how these camps were selected, assuming this is a clear rationale. 

Line 83: .'..still experiencing trauma' - While this is likely the case, please avoid sweeping statements that are beyond the research.Line 84: Please be more clear on the camp conditions meeting the Sphere standards and temperature is not a Sphere standard. This paragraph should be revised and focused on camp conditions and population demographics. The high standard of living and accustomed to flush toilets is useful, but taking away electronics etc. is not relevant. 

Line 100: Who (what organization) was doing hygiene promotion? The approach should be described and is an obvious factor in beneficiaries hygiene activities. One camp needing to buy soap could also be a huge factor compared to 'in-kind' distributions. 

Line 118-122: These sentences are not relevant

Line 144: Some words required brainstorming - how can you be sure the beneficiaries understood the questions?   Line 193: So, someone describing 2 times washing their hands would be a non-doer? Or someone who described washing each of th3 5 time points, but ran out of soap would also be a non-doer? 

Line 210: Please state the statistics with p-value.

Results: Are the camps comparable? Was analysis conducted to determine they were similar?

Result tables should be better organized to highlight that results are by camp. 

Table 4: the only significant finding thus far is that non-doers state reasons why more than doers. Please explain or discuss. 

Line 252 Social Norms - results are presented differently. Please be consistent and give percentages over absolute numbers because its not clear what the same size is. 

Line 292 table: Doers say child very likely to get sick if they wash at the 5 critical times. Things like this are odd - some explanation should be given. 

Line 299: Please give stats. 

Line 319: God's will being a reason people get sick is a pretty large factor. 

Line 336: To me, it seemed like there were little similarities. No topic was significant in both camps correct?

Reviewer #3: This paper provides a practical assessment of the determinants of handwashing behaviour among internally displaced women who had a child under the age of five in two camps in the Kurdistan Region of Iraq . This is a topic for which there is a substantial need for research, and it is encouraging to see the authors taking on the challenge of contributing to this gap in evidence. Generally speaking, this is a very clear paper that I would recommend for publication with some revisions and additions.

The title of the does not represent the study population. The study team administered the questionnaires to “women who had a child under the age of five” but authors used “internally displaced people” in the title. Authors should change the study title to make it clear. The introduction could be a bit more comprehensive in terms of contextualizing the research based on existing evidence. The methods section is straightforward but should add some missing references. The results/discussion section is solid, though Table 2-8 in particular had 2 unnecessary column and should be removed from each table. The conclusion is quite vague and should be bolstered with more specific recommendations based on the results of this study.

Below I have provided a few points which should be addressed:

1. The introduction could be a bit more comprehensive in terms of contextualizing the research based on existing evidence.

2. In the abstract in line 38, the terminology "demographics" is used. It would be good to use “socio-demographics”.

3. Line 66-68: Authors should remove this "In turn, it is hoped that an increased understanding of behavioural determinants will allow humanitarians to design more acceptable and effective handwashing promotion programmes "

4. Line 82-83: The authors should provide reference for this “All participants had been exposed to extreme violence in the past three years and many were still experiencing trauma at the time of this research”.

5. Line 90-91: The authors should provide reference (s) for this “Many of the residents had come from urban or peri-urban areas and were used to a relatively high standard of living prior to the conflict.

6. The title of the does not represent the study population. The study team administered the questionnaires to “women who had a child under the age of five” (Line 149) but authors used “internally displaced people” in the title. Authors should change the title.

7. Line 173: “to maintain quality and identify any missing data” Authors should make it clear how they maintain the quality of data and what they actually had done it they could identify any missing data.

8. Table 2-8 had 2 unnecessary column and should be removed from each table.

9. The conclusion is quite vague and should be bolstered with more specific recommendations based on the results of this study.

6. PLOS authors have the option to publish the peer review history of their article (what does this mean?). If published, this will include your full peer review and any attached files.

Reviewer #1: No

Reviewer #2: No

Reviewer #3: No

---

## [Author Response · Author response to Decision Letter 0]

28 Jan 2020

We thank the reviewers for their feedback and feel that their recommendations have enhanced the manuscript. We have edited the manuscript to address the reviewers’ concerns and responded to each comment point-by-point below.

We have submitted a clean copy of the manuscript and one with tracked changes. 

Editor Comments:

Response: We have revised the formatting of the document to be more consistent ant with the PLOS ONE journalistic style. Specifically we have changed the section headers and the Supplementary information file names.

2. Please include additional information regarding the survey or questionnaire used in the study and ensure that you have provided sufficient details that others could replicate the analyses. For instance, if you developed a questionnaire as part of this study and it is not under a copyright more restrictive than CC-BY, please include a copy, in both the original language and English, as Supporting Information. Additionally, please include upon how many participants the pre-testing of the questionnaire occurred.

Response: There was only one research tool used for this work. This was already attached as a supplementary information file. This questionnaire follows the standard format for Barrier Analysis surveys of handwashing behaviour.

3. Please refer to any post-hoc corrections for multiple comparisons you made during your statistical analyses. If these were not performed please justify why.

Response: No post-hoc corrections were made for multiple comparisons. 

4. We note you have included a table to which you do not refer in the text of your manuscript. Please ensure that you refer to Table 7 in your text; if accepted, production will need this reference to link the reader to the Table.

Response: There was an error in the numbering of this table and this has now been corrected in the text.

5. Please include captions for your Supporting Information files at the end of your manuscript, and update any in-text citations to match accordingly. Please see our Supporting Information guidelines for more information: http://journals.plos.org/plosone/s/supporting-information

Response: We have changed the in text reference to this file and added full captions at the end of the manuscript. 

6. Your ethics statement must appear in the Methods section of your manuscript. If your ethics statement is written in any section besides the Methods, please move it to the Methods section and delete it from any other section. Please also ensure that your ethics statement is included in your manuscript, as the ethics section of your online submission will not be published alongside your manuscript.

Response: Details about our ethics and consent process are included in the Methods section already and have been removed from the end of the manuscript. 

Review 1 comments:

1. It is good to see a study on handwashing in humanitarian settings. The authors are well justified in their contention that this kind of research is needed.

Response: We thank the reviewer for their kind words. 

2. The major technical limitation of the study lies with the identification of doers and non-doers. The authors report that “The screening process used a combination of self-reported handwashing behaviour and proxy measures of handwashing behaviour (such as the observed presence of used soap at the handwashing facility).” Self-reported handwashing is notoriously overreported:

https://journals.lww.com/jhqonline/Abstract/2007/07000/Reliability_and_Validity_of_Hand_Hygiene_Measures.5.aspx

Even direct observation of handwashing can overestimate actual handwashing: http://www.ajtmh.org/content/journals/10.4269/ajtmh.2010.09-0763

Script-based covert recall may be an alternative: https://journals.plos.org/plosone/article?id=10.1371/journal.pone.0136445

In this setting, proxy measures themselves may be inaccurate, and soap may be more available in a humanitarian setting than it would be normally.

Therefore, the study does not convincingly measure handwashing in a valid manner, and this throws into doubt the validity of classification of study participants as doers and non-doers.

Response: We thank the reviewer for raising these limitations and agree with their concerns. We already highlighted the limitations of self-reported measures in the discussion (line 403) but we have revised this to make it clearer and some of the alternative measures they suggest. Note also that the purpose of this manuscript was to assess the BA method as a commonly used assessment technique, not necessarily as a method we endorse. Self-report is still the only behavioural measure included in standard BA surveys. 

3. “Perceived social norms - The perception that people important to an individual think that he/she should wash their hands with soap. Note that in theories such as the Focus Theory of Normative Conduct, this is the definition of injunctive norms. See: Cialdini, R.B.; Reno, R.R.; Kallgren, C.A. (1990). "A focus theory of normative conduct: Recycling the concept of norms to reduce littering in public places". Journal of Personality and Social Psychology. 58 (6): 1015–1026. doi:10.1037/0022-3514.58.6.1015.

Response: We recognise this terminology discrepancy and that there is substantial disagreement on terminology to describe behavioural determinants in general. Our use of the term ‘perceived social norms’ in this manuscript is based on the definition of terms laid out in the BA guide. We have added into the discussion that we feel this is a particularly narrow conception of norms and suggested that future users refer to broader literature on this subject. 

4. Wrong statistical methods. In Tables like Table 8, the outcome or dependent variable “How likely is it that your child will get diarrhoea in the coming three months?” is an ordinal variable (See https://en.wikipedia.org/wiki/Ordinal_data for a definition) with three ordered responses. Appropriate measures of correlation include Kendall’s Tau-B, see https://en.wikipedia.org/wiki/Ordinal_data#Bivariate_statistics. It is inappropriate and incorrect to calculate a p-value for each row of the table. One single p-value and measure of association should be calculated with a statistical test like Kendall’s Tau-B. The same applies to Table 6 and 7.

Response: We followed the statistical methods recommended by the Barrier Analysis. We agree with the reviewer that there are some important limitations of these. Rather than re-do the analysis post-hoc we have decided to describe these limitations in the discussion. 

5. In the absence of accompanying qualitative interview and observation data, it is difficult to draw any conclusions about why people do or do not wash their hands.

Response: We agree that there are limitations with using this method alone. However, we also saw merit in testing this method. In our case, this BA study was done alongside other qualitative work (including observation) which is going to be reported separately. We felt that if we are going to persuade humanitarians to use more time consuming and complex methods then we needed to know the value add of these in comparison with the BA survey which is the status-quo. 

MINOR SPELLING AND GRAMMAR ISSUES

1. Please do a spell-check e.g. there are three errors here on the first page: “This [research] was made possible by the generous support of the American people through the United States Agency for International Development (USAID). The contents are the responsibility of the study authors and do not necessarily reflect the views of USAID or the United States Government. A grant from the Office of U.S. Foreign Disaster Assistance was [received] by SW (award number AID-OFDA-G-16- 00270). Funder website: https://www.usaid.gov/who-we- are/organization/bureaus/bureau-democracy-conflict-and-humanitarian- assistance/office-us. The funders played no role in the study design, data collection, analysis, decision to publish or [preparation] of the manuscript.”

2. Line 289 – delete ‘of’ or reword that part of sentence

3. Line 388 – change subtler to more subtle

4. Line 397 – remove comma

Response: We have gone through and made these specific changes as well as doing a general edit. 

Reviewer 2 comments:

1. Thank you for this interesting work. Overall, this is well written and useful for the humanitarian sector. A good edit of the intro, methods, and results is needed; I thought the discussion was particularly well written and organized.

Response: We than the reviewers for their kind comments. WE have done a full read through to pick up on grammatical errors. 

2. Something that I felt was missing was the discussion of WASH activities in the camps and previous/current hygiene promotion. NGO presence and activities need to be explained. By not describing NGO activities, the assumption is that NGO activities, promotion or distribution of materials or cash/vouchers have no impact on actions - I believe that to be rather significant and must be addressed.

Response: We have added some information about the nature of hygiene promotion in the study site description. Our study did not seek to measure the impact of existing NGO activities in relation to Hygiene Promotion and as such the survey was only done at one time-point. So we cannot draw a conclusion about the effectiveness of these particular activities in this context. However, these types of approaches have been widely used elsewhere and have not been able to realise substantial changes in behaviour. 

3. Differences were mentioned briefly about the camps, but more analysis should be presented. If the different camps are actually similar, would combining the data sets provide a better opportunity for significant results.

Response: Although the camps were in the same geographical location they were quite different in terms of their populations, facilities, regulations, and the duration of displacement. The rationale for purposively selecting these camps was to see if behaviour different substantially in these two settings. Hence we have chosen not to combine the two sites. 

4. How the camps were selected from what I would assume to be dozens of camps needs to be explained.

Response: The camps were purposively selected. We have added this along with the criteria used into the manuscript. 

5. Within the results there were several interesting findings (significant or not) that would be against logical assumptions, i.e. non-doers thought washing their hands would make them safer compared to doers (or something along those lines). While not significant, some explanation should be given or rationalized, especially since it seemed to occur several times.

Response: We have now added a section on this in the discussion

6. Line 52-3: Strong statement. Please reference. Limited water availability or lack of soap could also be reasons. 

Response: This statement is backed up with reference in the following sentence. A few sentences further on we also discuss the lack of access to water and soap as a barrier. 

7. Line 64: Please be explicit about 'this research gap'

Response: Thanks for pointing out that this was unclear. WE have re-written this section to be more specific. 

8. Methods: Please explain how these camps were selected, assuming this is a clear rationale. 

Response: The camps were selected purposively based on a set of criteria. This has been added to the manuscript. 

9. Line 83: .'..still experiencing trauma' - While this is likely the case, please avoid sweeping statements that are beyond the research.

Response: We have clarified that this was based on reports from participants and camp management. 

10. Line 84: Please be more clear on the camp conditions meeting the Sphere standards and temperature is not a Sphere standard. This paragraph should be revised and focused on camp conditions and population demographics. The high standard of living and accustomed to flush toilets is useful, but taking away electronics etc. is not relevant. 

Response: We have chosen to leave in this section on temperature and intermittent electricity because we feel it does provide a useful understanding of context. Temperature factors are known to affect handwashing behaviour (with hot weather more likely to make people feel sticky and dirty and cue handwashing). 

11. Line 100: Who (what organization) was doing hygiene promotion? The approach should be described and is an obvious factor in beneficiaries hygiene activities. One camp needing to buy soap could also be a huge factor compared to 'in-kind' distributions. 

Response: We have added that hygiene promotion was done by ‘international and local non-government organisations (NGOs) in conjunction with hygiene promoters from the camp population’. We do not feel it is necessary to name the organisations working there at this time. The fact that one camp no longer provided hygiene kit distributions was one of our reasons for selecting it. Observationally we noted that there was no shortage of soap in this camp however we do see differences in the results based on perceived affordability of soap in this camp. 

12. Line 118-122: These sentences are not relevant

Response: We feel that this section is relevant as appraising the merits and limitations of the BA approach was a secondary aim of our work. 

13. Line 144: Some words required brainstorming - how can you be sure the beneficiaries understood the questions? 

Response: We have added that this was done through a focus group discussion. Although we didn’t describe this in depth, pour translation team came up with several possible terms initially. We then get members of the local population to define each of the terms in their own words and use them in a sentence. This helped us identify which terms would be most similar to the English version of the questionnaire. To be clear this was all done to arrive at the final Arabic and Kurdish questionnaire. A standard version of each question was used in the actual survey. 

14. Line 193: So, someone describing 2 times washing their hands would be a non-doer? Or someone who described washing each of th3 5 time points, but ran out of soap would also be a non-doer? 

Response: The questions asked in the screening are provided in full in the supplementary material. A person who could only state 2 occasions when hands should be washed (despite additional probing) would be classed as a non-doer. A person who could list more than 3 occasions for handwashing but who did not have soap and water at the handwashing facility was classified as a non-doer. Soap scarcity was not a major issue in this population, a larger barrier was hoarding soap or keeping it in the household so that it wouldn’t be stolen by others. 

15. Line 210: Please state the statistics with p-value.

Response: This has been added. 

16. Results: Are the camps comparable? Was analysis conducted to determine they were similar?

Response: As discussed in the paragraph beginning on line 372, the camps were intentionally different in terms of facilities and socio-demographic characteristics. However, we found quite a lot of similarities in their perceptions around handwashing behaviour. We also discuss key differences in this paragraph. 

17. Result tables should be better organized to highlight that results are by camp. 

Response: We have changed the formatting to making this clearer. 

18. Table 4: the only significant finding thus far is that non-doers state reasons why more than doers. Please explain or discuss. 

Response: There are several statistically significant findings, specifically:

• Non-Doers in C2 are more likely to think that there is no likelihood of their children getting diarrhoea in the next 3 months. 

• Doers in C1 being more likely to think that there is somewhat of a likelihood of getting diarrhoea in the next 6 months. 

• Non-doers in C1 being more likely to think that it is somewhat difficult to remember to wash hands at critical times. 

• Doers in C1 were more likely to think that their mothers would approve of their handwashing. 

• Non-doers in C1 were more likely to think that an advantage of handwashing was removing dirtiness from hands. 

Each of these is described in the text we have also added a section explaining some of the more surprising findings. 

19. Line 252 Social Norms - results are presented differently. Please be consistent and give percentages over absolute numbers because its not clear what the same size is. 

Response: Thanks for picking up on this we have now added percentages to this section also. 

20. Line 292 table: Doers say child very likely to get sick if they wash at the 5 critical times. Things like this are odd - some explanation should be given. 

Response: We have added a section in the discussion where we discuss some of these unexpected findings. 

21. Line 299: Please give stats. 

Response: We have added the p-vlaue for the religion question. All other stats were already included in the written description. 

22. Line 319: God's will being a reason people get sick is a pretty large factor. 

Response: We agree with the reviewer’s point on this. We feel that this one question is insufficient to fully understand this determinant as we have mentioned in the discussion. 

23. Line 336: To me, it seemed like there were little similarities. No topic was significant in both camps correct?

Response: We are a little unclear what the reviewer meant by this comment but the text on line 336 refers to a comparison between known determinants of handwashing in stable settings (this is what the Barrier Analysis was developed based on) and determinants that are either not accounted for in the BA survey or are not sufficiently accounted for in the BA survey but may be more important in crisis settings. 

Reviewer 3 comments: 

1. This paper provides a practical assessment of the determinants of handwashing behaviour among internally displaced women who had a child under the age of five in two camps in the Kurdistan Region of Iraq. This is a topic for which there is a substantial need for research, and it is encouraging to see the authors taking on the challenge of contributing to this gap in evidence. Generally speaking, this is a very clear paper that I would recommend for publication with some revisions and additions.

Response: We thank the authors for their kind comments. 

2. The title of the does not represent the study population. The study team administered the questionnaires to “women who had a child under the age of five” but authors used “internally displaced people” in the title. Authors should change the study title to make it clear. 

Response: We have changed the title to ‘The determinants of handwashing behaviour among internally displaced women in two camps in the Kurdistan Region of Iraq’

3. The introduction could be a bit more comprehensive in terms of contextualizing the research based on existing evidence. 

Response: We have added some additional citations to the introduction. We have also added a section on determinant assessment in stable settings as a point of comparison. If there are additional further points the reviewer feels we should address in this section, then we would be willing to take this more specific feedback on board. 

4. The methods section is straightforward but should add some missing references. 

Response: We are not really sure what the reviewer was referring to here. We have already referenced our main method and included links to other similar approaches. 

5. The results/discussion section is solid, though Table 2-8 in particular had 2 unnecessary column and should be removed from each table. 

Response: It is not clear from the reviewer’s comment which columns they feel are unnecessary. We have not made any changes but are happy to do so with more clarity. 

6. The conclusion is quite vague and should be bolstered with more specific recommendations based on the results of this study.

Response: We believe we have discussed the findings in detail. Indeed, other reviewers felt that our discussion was the strongest part of our manuscript. Specifically we already provide detail on the key opportunities for people working in this context and link this to broader literature. 

7. In the abstract in line 38, the terminology "demographics" is used. It would be good to use “socio-demographics”.

Response: this has been changed

8. Line 66-68: Authors should remove this "In turn, it is hoped that an increased understanding of behavioural determinants will allow humanitarians to design more acceptable and effective handwashing promotion programmes "

Response: This has been removed.

9. Line 82-83: The authors should provide reference for this “All participants had been exposed to extreme violence in the past three years and many were still experiencing trauma at the time of this research”.

Response: We cannot provide a reference for this. This is based on our work in these settings. We have clarified this in the text. 

24. Line 90-91: The authors should provide reference (s) for this “Many of the residents had come from urban or peri-urban areas and were used to a relatively high standard of living prior to the conflict.

Response: Again we do not have a reference for this nor do we feel it is necessary. As stated these IDPs came from Mosul and Sinjar respectively. Both were bustling, modern metropolises prior to the conflict. 

25. Line 173: “to maintain quality and identify any missing data” Authors should make it clear how they maintain the quality of data and what they actually had done it they could identify any missing data.

Response: This has been added. 

We will be happy to provide any additional clarifications or edits as necessary.

Yours sincerely,

Sian White

Corresponding Author

London School of Hygiene and Tropical Medicine

---

## [Decision Letter · Decision Letter 1]

31 Mar 2020

The determinants of handwashing behaviour among internally displaced women in two camps in the Kurdistan Region of Iraq

PONE-D-19-25323R1

Dear Dr. White,

We are pleased to inform you that your manuscript has been judged scientifically suitable for publication and will be formally accepted for publication once it complies with all outstanding technical requirements.

With kind regards,

Ginny Moore

Academic Editor

PLOS ONE

Additional Editor Comments (optional):

You will see that Reviewer #2 has made a couple of minor suggestions that you may wish to take into consideration if you are required to make any technical amendments to your manuscript. Please also clarify the result presented in lines 257-8 (that when citing positive consequences of handwashing there were no significant differences between groups) as Table 4 (camp 1) suggests otherwise (to get rid of dirtiness p=0.042*).  

Reviewers' comments:

Reviewer's Responses to Questions

**Comments to the Author**

1. If the authors have adequately addressed your comments raised in a previous round of review and you feel that this manuscript is now acceptable for publication, you may indicate that here to bypass the “Comments to the Author” section, enter your conflict of interest statement in the “Confidential to Editor” section, and submit your "Accept" recommendation.

Reviewer #2: All comments have been addressed

Reviewer #3: All comments have been addressed

2. Is the manuscript technically sound, and do the data support the conclusions?

Reviewer #2: Yes

Reviewer #3: Yes

3. Has the statistical analysis been performed appropriately and rigorously? 

Reviewer #2: Yes

Reviewer #3: Yes

4. Have the authors made all data underlying the findings in their manuscript fully available?

Reviewer #2: Yes

Reviewer #3: Yes

5. Is the manuscript presented in an intelligible fashion and written in standard English?

Reviewer #2: Yes

Reviewer #3: Yes

6. Review Comments to the Author

Reviewer #2: I found the paper to be very well written - especially the discussion and conclusion. Thank you for your hard work.

A few comments:

The objective is a bit unclear. Is it to understand the barriers for handwashing or to compare the Barrier Analysis with other tools? It seems to flip back and forth in the intro/methods.

Line 220: Why is n=156? Wouldn’t it be 180?

Line 272: Please explain: “non-doers were 18% more likely than doers to report a lack of 273 negative consequences (doers = 80%, non-doers= 98%, p=0.008)”

In the results, the descriptions seem to go from ‘doers and non-doers’ – to more then more general ‘people’ or C1 and C2 – obviously those are different groups, and reporting 18% of XXX can mean very different things depending on the specific description. Please be aware that can be difficult for the reader.

Line 447: There are 2 periods.

Reviewer #3: Authors have adequately addressed my comments and I would like to recommend this paper for publication

7. PLOS authors have the option to publish the peer review history of their article (what does this mean?). If published, this will include your full peer review and any attached files.

Reviewer #2: No

Reviewer #3: No

---

## [Editor Report · Acceptance letter]

30 Apr 2020

PONE-D-19-25323R1 

The determinants of handwashing behaviour among internally displaced women in two camps in the Kurdistan Region of Iraq 

Dear Dr. White:

I am pleased to inform you that your manuscript has been deemed suitable for publication in PLOS ONE. Congratulations! Your manuscript is now with our production department. 

With kind regards,

on behalf of

Dr. Ginny Moore 

Academic Editor

PLOS ONE